# Performance Restoration of Chemically Recycled Carbon Fibres Through Surface Modification with Sizing

**DOI:** 10.3390/polym17010033

**Published:** 2024-12-26

**Authors:** Dionisis Semitekolos, Sofia Terzopoulou, Silvia Zecchi, Dimitrios Marinis, Ergina Farsari, Eleftherios Amanatides, Marcin Sajdak, Szymon Sobek, Weronika Smok, Tomasz Tański, Sebastian Werle, Alberto Tagliaferro, Costas Charitidis

**Affiliations:** 1Research Lab of Advanced, Composite, Nano-Materials and Nanotechnology (R-NanoLab), School of Chemical Engineering, National Technical University of Athens, 9 Heroon Polytechniou, GR-15773 Athens, Greece; 2Department of Applied Science and Technology, Polytechnic University of Turin, Corso Duca degli Abruzzi 24, 10129 Turin, Italy; 3Plasma Technology Laboratory, Department of Chemical Engineering, University of Patras, GR-26504 Patra, Greece; 4Department of Air Protection, Faculty of Energy and Environmental Engineering, Silesian University of Technology, 44-100 Gliwice, Poland; 5Department of Heating, Ventillation, and Dust Removal Technology, Faculty of Energy and Environmental Engineering, Silesian University of Technology, 44-100 Gliwice, Poland; 6Department of Engineering Materials and Biomaterials, Faculty of Mechanical Engineering, Silesian University of Technology, 44-100 Gliwice, Poland; 7Department of Thermal Technology, Faculty of Energy and Environmental Engineering, Silesian University of Technology, 44-100 Gliwice, Poland; 8Faculty of Science, OntarioTech University, 2000 Simcoe Street North, Oshawa, ON L1G0C5, Canada

**Keywords:** carbon fibre-reinforced polymers, composites, recycling, solvolysis, sizing, plasma-in-bubbles

## Abstract

The recycling of Carbon Fibre-Reinforced Polymers (CFRPs) is becoming increasingly crucial due to the growing demand for sustainability in high-performance industries such as automotive and aerospace. This study investigates the impact of two chemical recycling techniques, chemically assisted solvolysis and plasma-enhanced solvolysis, on the morphology and properties of carbon fibres (CFs) recovered from end-of-life automotive parts. In addition, the effects of fibre sizing are explored to enhance the performance of the recycled carbon fibres (rCFs). The surface morphology of the fibres was characterised using Scanning Electron Microscopy (SEM), and their structural integrity was assessed through Thermogravimetric Analysis (TGA) and Raman spectroscopy. An automatic analysis method based on optical microscopy images was also developed to quantify filament loss during the recycling process. Mechanical testing of single fibres and yarns showed that although rCFs from both recycling methods exhibited a ~20% reduction in tensile strength compared to reference fibres, the application of sizing significantly mitigated these effects (~10% reduction). X-ray Photoelectron Spectroscopy (XPS) further confirmed the introduction of functional oxygen-containing groups on the fibre surface, which improved fibre-matrix adhesion. Overall, the results demonstrate that plasma-enhanced solvolysis was more effective at fully decomposing the resin, while the subsequent application of sizing enhanced the mechanical performance of rCFs, restoring their properties closer to those of virgin fibres.

## 1. Introduction

CFs are among the most widely used reinforcing materials in the manufacture of Fibre-Reinforced Polymers (FRPs). CFs can be fabricated into forms such as fabric, woven, yarn (continuous), or chopped (short fibres), while thermoset polymers are commonly used as the matrix in CFRP manufacturing [1]. The most utilised CFs are polyacrylonitrile (PAN)-based, known for their ease of production and ability to retain mechanical properties under extreme conditions. Other fibres, such as aramid, boron, and basalt, are also used, but to a lesser extent compared to glass and carbon fibres. Natural fibres are also being explored, although their potential is currently limited, mainly due to poor compatibility between fibres (polar and hydrophilic properties) and the polymeric matrices available on the market (non-polar properties) [2,3].

In recent years, CFRPs have gained significant attention across industries such as aerospace, automotive, wind energy, leisure, and sports due to their outstanding chemical, physical, and mechanical properties [4,5]. These properties arise from the combination of CFs’ low density and high strength, which give CFRPs a lightweight nature, making them ideal for high-performance applications [6]. The polymer matrix also provides stability against compression, allowing CFRPs to be used in structures exposed to extreme conditions [7,8]. However, the growing use of CFRPs has led to increased production and, consequently, a rise in waste. Historically, end-of-life (EoL) CFRPs were disposed of through landfilling, incineration, or grinding into fillers—all of which pose environmental challenges. In addition, the high production cost of virgin CFs highlights the need for sustainable and economically viable recycling methods [9]. Several recycling techniques have been developed, but the process remains complex due to the heterogeneous nature of CFRPs, composed of a matrix and reinforcement. Therefore, selecting the right recycling method is critical. Among the existing methods, mechanical, thermal, and chemical recycling stand out [6,10].

Mechanical recycling involves grinding the composite into smaller pieces for reuse in CFRP manufacturing. Thermal recycling decomposes the polymer matrix at high temperatures, leaving the fibres intact for recovery [11]. Chemical recycling focuses on depolymerising the resin matrix to reclaim the fibres. Thermal and chemical recycling have more potential than mechanical processes because they can produce continuous fibres, whereas mechanical recycling typically yields chopped fibres, which are less useful for high-performance applications [12].

Even though technological advancements in CFRP recycling have made significant strides and remain an active area of research, several challenges persist. These include incomplete resin removal, structural damage to CFs caused by extreme processing conditions, and inadequate surface characteristics of CFs for proper resin adhesion, which is crucial for manufacturing high-value composite materials. To address these challenges, the present investigation focuses on two recently developed chemical recycling methods for CFRPs with thermoset epoxy resin as the matrix: solvent-based solvolysis, and plasma-enhanced solvolysis and the subsequent effect of sizing on the mechanical properties. Notably, chemical recycling (commonly referred to as solvolysis) has been identified as the method that causes the least damage to fibres during the recovery process [12].

Solvolysis involves the chemical decomposition of polymers into mono- and oligomers. However, this process is considered high-risk because it often requires the use of strong chemicals, elevated temperatures, and pressures, which, if not carefully controlled, could lead to hazardous conditions, including unwanted side reactions or incomplete resin degradation. To mitigate these risks, solvolysis at low temperatures—typically below 200 °C—and atmospheric pressure have been explored. Conducting the reaction under such mild conditions requires the use of acids, bases, or catalysts [10]. For instance, a study by Peng et al. [13] demonstrated the effectiveness of this approach by performing solvolysis at 200 °C under ambient pressure for 4 h using a polyethylene glycol/NaOH system. The experiment achieved a decomposition efficiency of 84.1–93%, with the reclaimed CFs exhibiting minimal resin residues, a slightly oxidised carbon structure, and a mildly reduced degree of graphitisation, as confirmed through various characterisation techniques. Furthermore, the tensile strength of the recovered CFs was preserved at 94–96% of the pristine fibres, indicating the success of this method. In this research, epoxy-based CFRPs are recycled using low-temperature, low-pressure solvent-based solvolysis, employing a 10% KOH solution in ethylene glycol (EG). The use of KOH offers several advantages over NaOH, including higher solubility in ethylene glycol, enhanced catalytic activity, and better control of reaction pathways. These properties allow for more efficient resin decomposition while reducing potential structural damage to the fibres. Additionally, KOH is less corrosive, which further contributes to preserving the integrity of the recycled fibres. The solvent is reused across multiple experiments to maximise efficiency and minimise waste. Plasma-enhanced solvolysis, on the other hand, represents a novel and promising technology that combines the benefits of chemically assisted solvolysis with plasma chemistry. This approach has been explored in only a few studies, making it an innovative method for CFRP recycling. In our previous work [12], the use of nitric acid as a solvent was proposed, augmented with nitrogen plasma, to generate a variety of reactive species. These reactive species facilitate the rapid degradation of the resin. Plasma-enhanced solvolysis has been shown to require significantly less time than conventional HNO_3_ solvolysis while reclaiming fibres with mechanical properties comparable to virgin CFs. This innovative recycling method is used in this study to produce recycled fibres and study the sizing effect on their properties.

The goal of recycling EoL CFRP parts is to reclaim high-quality recycled carbon fibres (rCFs) for valuable applications. However, rCFs often have inert surfaces, resulting in poor adhesion to the matrix and reduced interlaminar shear strength, which can lead to delamination or sudden failure without visible damage [8]. Achieving strong fibre-matrix adhesion is crucial; better bonding increases the energy required to separate the fibres, thus enhancing overall performance. Fibre sizing, the application of a polymeric coating to the fibres, is one way to improve fibre-matrix adhesion. Sizing enhances resistance to environmental degradation and mechanical stress during handling and transportation while improving the interface quality between the fibres and matrix. Common sizing agents include epoxy resins, polyurethane, and polyamide, with the choice depending on the application [7]. Recently, the incorporation of nanomaterials, such as carbon nanotubes (CNTs) or nanoparticles with diverse morphologies, into sizing agents has gained attention for their unique chemical and mechanical properties [14].

As CFRP recycling advances, new characterisation methods are needed to assess the quality and integrity of rCFs accurately. Conventional methods often struggle to evaluate fibre loss and degradation during recycling, limiting process optimization. Therefore, novel techniques are required to provide insights into the physical properties of rCFs, such as morphology, distribution, and breakage. To address this, a novel optical microscopy-based quantification method is developed in this work to assess filament loss that may occur during the recycling process by accurately quantifying the number of filaments. While the proposed method focuses on the post-recycling evaluation, it serves as a valuable starting point for the development of further methodologies aimed at real-time monitoring of fibre quality during the recycling process. Such advancements could enable optimised recycling procedures and the production of high-quality fibres, not only in lab or pilot-scale settings but also in industrial-scale applications.

In summary, this study investigates the effects of fibre sizing on the properties of rCFs recovered from two chemical recycling processes: chemically assisted solvolysis and plasma-assisted solvolysis, applied to EoL automotive parts. The novelty of this work lies in the continuous exploration of sizing as a means to restore and enhance the properties of rCFs, specifically in conjunction with a novel plasma-assisted solvolysis method. Furthermore, the development of a custom Python-based image analysis script for quantifying filament loss introduces a new tool for evaluating the efficiency of recycling processes and assessing filament quality with higher accuracy and reproducibility. The morphological characteristics of rCFs were analysed using scanning electron microscopy (SEM), and their structural stability post-recycling was evaluated through thermogravimetric analysis (TGA) and Raman spectroscopy. Additionally, an automated optical microscopy analysis was introduced to measure filament loss during recycling. Tensile tests were performed on both recycled and sized fibres (single fibres and yarns) to evaluate the effects of recycling and sizing on their mechanical properties.

## 2. Materials and Methods

### 2.1. Materials

EoL tubular parts from automotive, with dimensions O.D 6 cm, I.D 5.5 cm, and H 20 cm, were supplied by B&T composites (Florina, Greece) fabricated with Tenax^®^-E STS40 (Teijin, Tokyo, Japan) continuous fibre via filament winding. The physical and mechanical properties of those CFs are listed below in Table 1.

For chemical-assisted solvolysis, the following chemical substances were used: potassium hydroxide (KOH, wt/wt 10% solution, purity > 99.9%, Honeywell) in ethylene glycol (EG, purity > 99.5%, Fisher Chemical, Waltham, MA, USA), while for plasma-assisted solvolysis: acetone (2-propanone, wt/wt 99.8%, Fisher Chemical), nitric acid (HNO_3_, wt/wt 65%, Honeywell), hydrogen peroxide (H_2_O_2_, wt/wt 30%, Carlo Erba reagents, Milan, Italy), and nitrogen gas (N_2_, purity > 99.9%, EVOXA, Paris, France).

Hydrosize^®^ HP2-06 (Michelman, Aubange, Belgium) was the commercial polymeric coating used for the sizing of rCFs. Its properties are presented in Table 2. It is an anionic/nonionic phenoxy aqueous dispersion that acts as a sizing agent for fibres, enhancing their compatibility with each matrix, and promoting the mechanical performance of the final composite.

For the preparation of samples for optical microscopy and tensile tests, the SR1710/SD8822 structural epoxy system from Fibremax Composites (Volos, Greece) was used (Table 3). SR1710/SD8822 is a two-component epoxy system that cures at 25 °C for 24 h and post-cures at 40 °C for an additional 24 h.

Five different sample categories are studied in this manuscript, and for the reader’s convenience, they will be referenced according to Table 4.

### 2.2. Recycling of EoL Parts

#### 2.2.1. Chemical-Assisted Solvolysis

The solvolysis process was conducted in a 2 L unpressurised batch reactor equipped with a reflux condenser and a nitrogen supply. A 10% KOH solution in ethylene glycol (EG) was used as the solvent. The solvolysis reaction took place at 190 °C for a minimum duration of 6 h. The CF tubes were secured using a stainless-steel wire basket, which held the filaments in place to reduce entanglement during the process. The secure sample was subsequently introduced into a preheated 10% KOH solution in ethylene glycol at 60 °C (to facilitate easier dissolution of KOH in EG). The mixture was then heated to a temperature range of 190–195 °C. Periodically, the reactor was opened to monitor the process. After 6 h, the stainless steel wire basket containing the CF was removed from the reactor and allowed to drip. Once the CF had cooled to room temperature, it was cleaned using tap water with a small amount of surfactant. The CF was then left to dry. The final CF mass was obtained after drying at 105 °C. The remaining KOH solution in ethylene glycol was collected for reuse in the next process.

#### 2.2.2. Plasma-Assisted Solvolysis

Plasma-assisted solvolysis involves 5 discrete steps, namely, material pre-treatment, plasma-assisted solvolysis, rCF cleaning, liquid waste regeneration, and flue gas scrubbing. A detailed flow chart is presented in our previous work [15]. Initially, the CFRPs are treated in a 4 M HNO_3_ solution for matrix swelling and then enter the plasma reactor, where they are treated up to complete matrix dissolution. The flue gas produced during plasma-assisted solvolysis flows through a wet scrubber containing a dilute HNO_3_-H_2_O_2_ solution so that the emitted NO_x_ is partially converted to HNO_3_. When the HNO_3_ concentration of the scrubbing liquid reaches 4 to 6 M, it is collected and used for the CFRPs pre-treatment. When the plasma-assisted solvolysis is completed, the liquid waste is regenerated by adding small amounts of H_2_O_2_ and is reused in the next solvolysis cycle. The continuously recovered fibres are mechanically collected, washed with acetone, and dried naturally. Figure 1 illustrates the plasma-in-bubbles reactor set-up. This type of plasma was chosen as it leads to the production of active species close to the composite surface, while at the same time the plasma-induced shockwaves inside the liquid favour the mass transport of resin fragments from the solid to the liquid phase [16]. The reactor consists of a 2 L glass container where CFRPs and concentrated (65% wt/wt) HNO_3_ are placed. The vessel is positioned on a stainless steel plate, which is the grounded electrode of the reactor. The powered electrode is a stainless-steel tube of ¼ in diameter that is immersed in the solution and through which the gas enters the liquid and produces bubbles. The electrode is powered by a high-frequency generator (30 kHz signal generator IGBT143, Martignoni Elettrotecnica, Vestone, Italy) through a voltage amplifier (IGBT163, Martignoni Elettrotecnica). A high-voltage 1000:1 passive probe (P6015A, LeCroy, Chestnut Ridge, NY, USA) is adjusted on the power line to record the applied voltage, while the current flow of the system is calculated by recording the voltage drop (100:1 passive voltage probe, Hameg HZ53, Hameg Instruments, Mainz, Germany) across a 6.5 Ω resistance attached between the stainless-steel plate and the ground. Recording the applied voltage and the current flow allows the calculation of the process power consumption [16]. In this work, the N_2_ flow rate, the HNO_3_ solution volume, and the power consumption were kept equal to 4 Lmin^−1^, 1.2 L, and 250 W correspondingly so that the temperature of the solution is stabilised to ~80 °C about 10 min after plasma ignition. The total time required for the complete matrix dissolution and the recovery of CFs was about 5 h.

The energy and cost efficiency of the two recycling methods—chemically assisted solvolysis and plasma-assisted solvolysis—are critical factors for industrial scalability. Chemically assisted solvolysis operates under relatively mild conditions, such as low temperature and atmospheric pressure, resulting in lower energy consumption compared to high-temperature methods like pyrolysis. The use of a 10% potassium hydroxide solution in ethylene glycol further optimises costs through solvent reuse, though longer reaction times and solvent recovery processes remain challenges for scaling. Plasma-assisted solvolysis offers faster reaction times at approximately 80 °C, significantly reducing processing duration while ensuring effective resin removal. Despite its higher energy demands due to the plasma system, the process benefits from efficient resource utilisation, as liquid waste streams and flue gases can be recycled within the system. While both methods are at a low technology readiness level (TRLs 3–5), their laboratory-scale energy profiles can provide valuable information for optimising cost-effectiveness and resource efficiency, facilitating their development toward industrial adoption.

### 2.3. Fibre Treatment

Solvolyzed CFs tend to exhibit an inactive surface, which results in composites with reduced mechanical performance. This decline is primarily attributed to the recycling process, which disrupts the alignment of the fibre tow and strips functional groups from the surface. In cases where the recycling process is particularly intense or involves harmful catalysts, the structural integrity of the fibres can also be compromised, although this can be monitored and mitigated. The lack of fibre-matrix covalent bonds weakens the adhesion between the fibres and the matrix. To improve these properties, a commonly employed method involves the application of polymeric coatings on the fibres, known as sizing. Sizing’s role is to protect the fibres during processing, to enhance the adhesion between the fibre and the matrix, as well as to aid the fibres to regain the mechanical properties they possess as virgin CFs. These solutions consist of a film former, lubricants, and additives, such as coupling agents and/or additives dispersed in water [17].

In this work, sizing is applied using a pilot-scale continuous sizing line (Figure 2). The line consists of seven serially installed elements, which are briefly introduced here, with full details available in our previous work [14]. The process begins with the let-off tension creel, which feeds the fibre into a furnace while maintaining the required tension in coordination with the take-up winder. The furnace is used (if needed) to remove any resin residues from the recycling process by adjusting the temperature between 300 °C and 600 °C, depending on the specific requirements. Afterwards, fibre passes through the sizing bath, where a coating solution is applied via bath rollers, and the excess solution is removed by squeeze rollers to ensure uniform coverage. The fibre then moves into the drying furnace, where solvent evaporation and coating solidification occur. The process is controlled by a feed roller system, with speeds ranging from 0.2 to 2 m/min, depending on production needs. At the end of the line, the fibre is collected by a mechanical traverse system and take-up winder, powered by a constant-torque motor.

Ideally, recycled CFs should remain continuous. However, depending on the recycling process and handling, the fibres may be cut randomly. When this occurs, the fibres must be rejoined using pneumatic splicing. This method involves overlapping the separated yarn ends and bonding them with turbulent air, creating a strong connection [18], using a splicer (Airbond, Pontypool, UK). In the early stages of process development, splicing was necessary to achieve continuous yarns. However, as the process was optimised, no splicing was needed. Once the continuous yarn is reclaimed, the spool is mounted on the sizing line. The sizing bath is filled with the commercial sizing solution Hydrosize^®^ HP2-06 (Michelman, Aubange, Belgium), with a solid content of 1%, as described in Section 2.1. The hydrasize sizing agent used here is a phenoxy dispersion in an amine emulsifier. Phenoxy resins in a mixture with amines will generate a hydroxyl group in the polymeric chain through the interaction of the phenoxy ester or ether with the amine group. The hydroxyl group will promote the covalent bonding of the sizing polymer to the rCF surface through either dehydration or esterification reactions with -OH or -COOH groups that already exist in the rCF surface. In addition, the hydroxyl group will enhance h-bonding interactions between the sizing polymer and rCF surface, thus leading to a stronger attachment.

### 2.4. Optical Microscopy Analysis Preparation

Optical microscopy is a characterisation technique that requires meticulous sample preparation with a flawless surface finish. Any imperfections, surface texture, or contaminants can significantly impact the accuracy of the analysis. Therefore, it is essential to prepare samples with a high-quality surface to prevent the microscope from capturing any irregularities that could interfere with the results.

The samples prepared for this research were aligned, enclosed in disc-shaped moulds, and impregnated with epoxy resin. To ensure that the fibres remained stretched and properly aligned throughout the curing period before placing them into the moulds, fibres were kept under tension on a metallic plate, and resin droplets were carefully applied using a pipette. The impregnated samples were then placed in an oven at 50 °C for 24 h. After curing, each specimen was cut into 1 cm pieces and embedded perpendicularly in a 30 mm diameter cylindrical mould. The fibres were held vertically with a small clamp and placed at the bottom of the mould, ensuring they stood upright and perpendicular to the surface. Each mould contained five samples, and additional resin was poured over them. The same curing procedure was followed, and after removal from the moulds, the specimens were ground and polished as detailed in Table 5 and Table 6.

### 2.5. Characterization Methods

To evaluate the effectiveness and impact of each recycling method on the fibre surface, samples were examined using SEM (TM3030Plus Tabletop Microscope, HITACHI, Tokyo, Japan) at magnifications up to ×1000. By scanning the surface of the samples, it was possible to identify the amount of residual resin remaining after the recycling process and to assess any potential surface damage. Five different samples per case were analysed, scanning the entire surface area of each and capturing representative images, which are presented in the Results section.

TGA (NETZSCH Proteus Thermal Analysis 8.0.2, Selb, Germany) was used to evaluate total mass loss at a specific temperature range, which is an indication of resin residue after the completion of the recycling process, as well as a quantitative determination of the polymeric coating that has been added to the recycled fibres. Furthermore, the structural integrity of the fibres can also be identified through this method, as any detection of significant weight loss would indicate potential fibre structural damage. TGA analysis was conducted in a nitrogen (N_2_) atmosphere with a flow rate of 50 mL/min, heating the samples to 900 °C at a rate of 10 °C/min. Further investigation on the structural behaviour was performed by Raman spectroscopy. The Raman spectra of fibres were collected using a Renishaw InVia (H43662 model, Gloucestershire, UK) equipped with a laser line emitting at a wavelength of 785 nm and a 50× objective lens. Raman spectra were recorded in the range from 500 cm^−1^ to 4000 cm^−1^, and the region between 500 cm^−1^ and 3000 cm^−1^ was analysed using home-made software compiled in MATLAB^®^ (version R2020a), following a procedure reported by Tagliaferro et al. [19].

XPS measurements were performed to identify the surface functional groups of carbon fibres at three different stages: Ref_CF, Pl_rCF, and Sized_Pl_rCF. This technique, with its shallow sampling depth of a few atomic layers, provides both qualitative and quantitative insights into surface modifications. The measurements were conducted using a UHV Prevac spectrophotometer (Rogow, Poland) equipped with a VG Scienta XM 780 monochromator and an Al Kα radiation source (1486.6 eV). High-resolution (HR) XPS spectra were recorded at a pass energy of 50 eV, with an energy step size of 50–100 meV. The binding energy scale was calibrated by setting the binding energy of the aliphatic carbon C1s peak to 284.8 eV. All spectra were analysed using CasaXPS Version 2.3.23 PR1 software. Due to limited availability, this method was applied specifically to Ref_CF, Pl_rCF, and Sized_Pl_rCF, as they demonstrated the best results from SEM and TGA analysis.

An Olympus BX53M microscope (Olympus, Tokyo, Japan) with a brightfield observation method and a camera resolution of 5760 × 3600 pixels was used for the inspection of the polished specimens described in Section 2.4. For image captioning and analysis, the Olympus Stream Motion software v 2.5.3 was utilized. Additionally, the Panorama feature was employed to combine multiple images into one, simplifying the analysis of the samples.

A tensile testing machine (WD100 TE machine, Jinan Testing Equipment IE Corporation, Jinan, China) was used to measure the tensile strength of carbon fibres, according to ASTM D4018 [20]. The tensile test specimens were impregnated with the SR1710/SD8822 (as presented in Section 2.1) and consolidated. The testing parameters of the machine were set as follows: the tensile rate was 2 mm/min with a load cell of 5 kN. The specimens were untabbed, requiring a distance of 150 mm between the grips. Five specimens of each kind of fibre were tested. To calculate tensile strength (MPa), the following formulas were used:(1)$$UTS=A\times P\times \rho_{f}/MUL$$
where:UTS: Ultimate Tensile Strength (MPa)P: maximum load measured in tensile test, N;*ρ*_f_: fibre density, g/m^3^;MUL: fibre mass per unit length, g/m; andA: unit conversion factor (1 if load in N).
(2)$$MUL=W_{1}/L$$
where:MUL: mass per unit length, g/m;W_1_: mass of the specimen, g; andL: length of the specimen, m.

For the single fibre tensile test, individual filaments were carefully selected from a fibre bundle. Each filament was mounted on a paper frame with a 25 × 10^−3^ m gauge length, and the top and bottom ends were securely glued to the frame. The entire frame was then positioned in the grips of the tensile testing machine with a load cell of 5 N. Before initiating the test, the edges of the paper frame were cut with scissors to release the fibre. The test was conducted at a constant crosshead speed of 1 mm/min until the fibre reached the point of breakage. Ten specimens of each kind of fibre were tested. To calculate the tensile strength (MPa), the following classic formula for normal stresses was used:(3)UTS=P/A
where:UTS: Ultimate tensile strength (MPa)P: maximum load measured in the tensile test, N; andA: area of single fibre cross-section, mm^2^.

## 3. Results and Discussion

### 3.1. rCFs Surface Morphology Assessment

#### Scanning Electron Microscopy

Figure 3 presents the SEM images of Ref_CF, along with Pl_rCF and Ch_rCF. Ref_CFs (Figure 3a) exhibit a smooth and ridged surface, as expected from untreated fibres. The EDS analysis, performed in the spot highlighted with yellow in all images, revealed a carbon content of 96% (Figure 3b) and 4% of oxygen. This small content of oxygen can be explained by partial oxidation on the carbon fibre surface or by the application of organic sizing agents [21]. In contrast, the Ch_rCF (Figure 3c) shows the presence of a few resin residues on the surface, though no filament damage was observed during the visual inspection. These remnants are likely a result of incomplete removal during the chemical recycling process. The EDS analysis of the selected spot revealed 84% carbon content and 16% oxygen (Figure 3d), which is expected since the area analysed is likely resin-rich. This lower carbon percentage suggests that the EDS spot included regions where resin still clung to the fibre surface, which might slightly impact the mechanical properties of the fibres post-recycling. For Pl_rCFs, SEM images (Figure 3e) show a clean surface free from visible resin residues and no observable damage to the filament structure. EDS analysis reveals a carbon content of 93.5% and an oxygen content of 6.5% (Figure 3f). This oxygen presence can be attributed to plasma treatment, which not only assists in removing residual polymer matrix but also introduces oxygen-containing functional groups, such as carboxyl (-COOH) groups, onto the fibre surface. These functional groups enhance the surface energy of the fibres, promoting stronger adhesion to the resin matrix and improving the overall bonding performance in composite applications. However, sizing is still necessary since these groups alone may not provide sufficient durability and mechanical stability during composite manufacturing. Sizing plays a complementary role by forming a protective polymeric layer on the fibre surface, which not only enhances the interfacial bonding but also protects the fibres during handling and processing. This surface modification is further confirmed by XPS analysis, as discussed in Section 3.3.3, which demonstrates the presence of oxygen-containing groups on the surface of Pl_rCF, reinforcing the impact of this process on fibre surface chemistry.

Following the recycling process, the yarns were subsequently subjected to sizing, followed by SEM analysis. On Sized_Ch_rCFs (Figure 3g), a few areas with resin residues were still visible. The passage of these fibres through the 1st furnace of the pilot scale sizing line prior to sizing facilitated the partial removal of the resin residues, though it did not fully eliminate them. As a result, the EDS analysis showed a carbon content of 93% and 7% oxygen (Figure 3h). In contrast, Sized_Pl_rCFs (Figure 3i) exhibited a smoother surface morphology, reflecting better surface preparation and more effective resin removal during plasma treatment. This cleaner surface facilitated the adhesion of the sizing material. EDS analysis of these fibres showed a carbon content of 93% and an oxygen content of 7% (Figure 3j). A slightly higher oxygen level is expected, as plasma treatment introduces oxygen-containing functional groups on the fibre surface, and the application of sizing further contributes to the oxygen detected. The sizing process involves the use of polymeric coatings, which include oxygenated components, contributing to the observed oxygen levels. Quantitative data from the EDS analysis are provided in the Appendix A.

### 3.2. rCFs Structural Characteristics

#### 3.2.1. Thermogravimetric Analysis

The TGA graph (Figure 4) presents the thermal behaviour of Ref_CF, Sized_Ch_rCF, and Sized_Pl_rCF up to 900 °C. For Ref_CF, a minimal mass loss of 0.5% is observed, indicating that these fibres remain stable and unaffected by thermal stress within this temperature range. This confirms that they retain their structural integrity, as expected for untreated carbon fibres [22]. If the recycling process does not compromise the fibres’ properties, similar thermal behaviour would be expected in the recycled and sized samples. For Sized_Pl_rCFs, a mass loss of approximately 2% is observed. There is a distinct drop in the graph around 350 °C, which likely corresponds to the decomposition of the sizing material applied after the recycling process. This temperature range is consistent with the thermal degradation of common sizing agents, which typically contain polymeric materials. Despite this loss, the lack of significant mass reduction beyond this point indicates that the fibres maintain their structural integrity after plasma treatment. In the case of Sized_Ch_rCFs, a similar trend is observed, though the mass loss is steeper, with a total loss of 4.59%. The drop around 350 °C is attributed to both the decomposition of residual resin after the recycling process and the degradation of the sizing material, further confirming the SEM findings.

#### 3.2.2. Raman Spectroscopy

The Raman spectrum (Figure 5) of the reference fibre shows a D and a G peak centred at 1334 cm^−1^ and 1592 cm^−1^, respectively. The G peak, associated with the in-plane vibrations of sp^2^ hybridised carbon atoms, represents the graphitic structure, while the D peak indicates structural defects or disorder within the carbon structure. Additionally, the shoulder observed at 1186 cm^−1^ could correspond to structural defectiveness as reported by Shimoidara et al. [23] related to sp^2^ hybridised carbon clusters and bond angle disorder. These defects may result from both (i) an oxidation process and (ii) an applied sizing layer on the fibre surface, intended to improve fibre-matrix adhesion in composite applications.

The fibre recycled through plasma-assisted solvolysis displays a similar pattern with a D and a G peak centred at 1331 cm^−1^ and 1597 cm^−1^, respectively, together with a shoulder at 1119 cm^−1^, suggesting the preservation of original features.

In contrast, the fibre recycled with traditional solvolysis exhibits a greater level of structural complexity. The D peak shifted at 1353 cm^−1^, the G peak at 1593 cm^−1^, and the other two minor components were centred at 1122 and 1191 cm^−1^. Further signals at 1833 cm^−1^ and 3268 cm^−1^ suggest the presence of functional groups such as C=O and OH, suggesting the residual presence of epoxy resin together with an advanced oxidation of the surface. The great intensity of the D peak supports the great impact of the solvolysis process on the fibre quality.

The comparison of the D and G band intensities across the samples highlights the differing effects of the recycling processes on fibre structure. The reference spectrum is similar to that of the plasma-recycled fibres. In contrast, the Raman spectrum of chemically recycled fibres suggests the presence of resin residues (as also observed by SEM and TGA). Solvolysis’ impact will be further assessed through mechanical testing; a deterioration in mechanical properties would indicate that the solvolysis process affects fibre integrity, while stable mechanical performance would imply that the spectrum is primarily influenced by residual resin rather than structural damage.

### 3.3. Automated Analysis of Optical Microscopy Images

#### 3.3.1. Optical Microscopy Analysis

Figure 6 presents the different types of materials examined in this research: Ref_CF, Pl_rCF, and Ch_rCF. The grey regions correspond to the epoxy resin used in the moulds for sample preparation. The black holes represent air pockets, while the white specks depict carbon fibres. Both Pl_rCF and Ch_rCF samples show black discolouration surrounding the fibres, likely due to insufficient polishing time. This issue arises as the polishing machine applies pressure through a pin that descends and presses on the centre of the disc sample, leading to less polished areas near the outer edges. To address this, all three samples were polished for an additional 10 min using a 1 μm suspension. This was sufficient for Ref_rCFs, but Pl_rCFs and Ch_rCFs required an extra 7 min of polishing. After the extended polishing, all samples appeared appropriate for analysis. Although some dark spots remained on Pl_rCF and Ch_rCF samples, the surface quality was significantly improved.

#### 3.3.2. Image Analysis

This analysis aimed to determine whether the recycling process leads to a reduction in filament count and to compare the efficiency of three different validation methods: first, the segmentation tool within the Olympus streamt motion software v 2.5.3; second, a particle-counting plug-in within the ImageJ software v1.54m; and finally, a new Python script version v1 developed from scratch for fibre counting. The scripts can be found online [24]. More detailed information about each analysis is introduced below. Five samples per case were analysed, and the filament count is presented in Table 7. Results from each method are analysed below. Besides the Olympus software, which is clearly inaccurate, the results from both ImageJ and the Python script show a loss of approximately 1500 filaments during the recycling and sizing process. This corresponds to a filament loss of about 6% when compared to the original reference fibre, which had an expected filament count of 24,000. Since no structural damage was observed in the previous analyses, this reduction in filament count could still impact the mechanical properties of the yarns, potentially weakening their overall performance.

##### Olympus Software Analysis

The fibre analysis using the Olympus software began with importing optical microscopy images captured with the Olympus microscope (Figure 7a). Image enhancement tools were applied, adjusting brightness and contrast and optimising the visibility of the fibres against the background. Semi-automated methods provided by the software were then employed to identify and count the fibres. These methods utilise algorithms to segment fibres based on their colour, intensity, and texture. Once the fibres were accurately identified, their count was calculated. After considering that the expected filament count for the Ref_CF sample should be close to 24k, the Olympus software provided a much lower count of around 15k, indicating a significant error. Upon closer examination of the post-analysis images (Figure 7a), it became clear that the software struggles to differentiate among fibres that are adjacent to each other or broken fibres that lack a consistent circular shape. Despite this limitation, the Olympus software remains the fastest option, offering immediate analysis post-image capture. Nevertheless, it tends to perform poorly in images with dense or overlapping fibres, limiting its reliability in more complex cases.

##### ImageJ Analysis

Analysing images with ImageJ for fibre quantification involves a series of sequential steps. First, the selected image is imported into the software and converted to an 8-bit format, turning it into grayscale. The ‘Brightness/Contrast’ tool is then applied, accessible through ‘Image’ followed by ‘Adjust’, allowing the visibility of the fibres to be enhanced. Typically, contrast is increased while brightness is reduced to make the fibres appear white against a black background, which makes them easier to distinguish. Next, the image undergoes binarization through the ‘Make Binary’ function under ‘Process’, converting it into a black-and-white format based on a threshold value. To accurately segment overlapping or adjacent fibres, the ‘Watershed’ algorithm is applied. Once the image is properly processed, the ‘Analyse Particles’ function is used to set parameters such as circularity and size, helping to filter out artefacts or irrelevant features. Finally, the software computes the number of fibres in the image (Figure 7c) based on the specified criteria, providing quantitative data on fibre distribution within the resin cross-section. ImageJ is a highly reliable method, offering a systematic, step-by-step process that guarantees precision. However, it demands considerable time, as each image must be manually and individually analysed. The ‘Watershed’ algorithm is especially efficient in separating overlapping or adjacent fibres, enhancing the accuracy of the results, and ensuring consistent performance.

##### Python Script

The developed script (contoursScript.py) is an automated tool designed to quantify fibre counts from optical microscopy images, offering a more accurate and reliable approach compared to alternative developments [24]. After importing the necessary Python libraries and loading the images from a designated folder, the script converts the images to grayscale and applies a binary threshold to distinguish fibres from the background. Contours representing the fibres are detected, outlined, and counted. The results are saved both in the processed image file name (Figure 7b) and an Excel worksheet. This method successfully yields fibre counts close to the expected value (~24,000), ensuring reproducible and systematic analysis of fibre distribution within the resin cross-section. The contoursScript.py provides an efficient solution for evaluating the damage or degradation of yarns in composite recycling by offering precise fibre quantification. A full analysis, along with the scripts, can be found in [24].

#### 3.3.3. XPS

XPS analysis was conducted to identify the surficial functional groups of carbon fibres at three different stages: Ref_CF, Pl_rCF, and Sized_Pl_rCF. Table 8 presents the elemental composition for each fibre type. In the Ref_CF sample, carbon remains the predominant element, constituting 83.4% of the total composition, followed by oxygen at 14.9%. Minor elements such as silicon and chlorine were detected at lower levels (0.9% and 0.4%, respectively). Trace amounts of contaminations such as silicon and chlorine were also observed in similar XPS studies of virgin fibres. Thus, it is probable that they are picked up during the CF production process [25]. Upon recycling, the carbon content shows a noticeable reduction to 75.7%, while oxygen increases to 18.5%, reflecting the surface changes caused by the recycling process. Nitrogen, absent from the Ref_CF sample, appears in the Pl_rCF sample at 4.0%. HNO_3_ and plasma in N_2_ bubbles processing that is used for the recycling of this sample will produce a number of extremely reactive N- and O-containing species such as NO_2_^+^, OH•, H_2_O•, NO_2_•, H_2_O_2_, O^+^, OH^−^, O_3_, NO_3_^−^, O_2_^−^, NO_2_^−^ [15]. These moieties, except for dissociating the polymer matrix, can be also grafted to the CF surface, and this is estimated as the main reason for the increase in N and O content in the Pl_rCF. Trace amounts of silicon and chlorine that were observed in Ref_CF are also present in Pl_rCF together with traces of calcium, which probably results from the sputtering of soda lime glass that is used as dielectric in the plasma-powered electrode.

For the Sized_Pl_rCF sample, carbon content increases slightly to 79.0%, while oxygen shows a minimal increase to 19.3%. The elemental composition of Sized_Pl_rCF closely resembles that of Ref_CF. This suggests that the sizing process restores the surface composition of the recycled carbon fibres to a state very similar to commercial fibres in terms of atomic percentage, particularly for carbon and oxygen, thus enhancing the surface characteristics to mimic those of commercially available fibres.

XPS analysis of the C1s peaks further highlights the changes in surface chemistry across the three carbon fibre samples. Table 9 presents the detailed decomposition of the C1s peaks, revealing the abundance of the various functional groups.

Figure 8, Figure 9 and Figure 10 represent the high-resolution spectra for Ref_CF, Pl_rCF and Sized_Pl_rCF. The colored curves represent the deconvoluted peaks corresponding to different chemical components. The gray circles indicate the experimental data points, while the smooth curves represent the fitted spectra. For Ref_CF (Figure 8), the predominant peak corresponds to C-C/C-H bonding at 285.0 eV, accounting for 67.2% of the total carbon composition. This high concentration of C-C/C-H bonds is consistent with the graphitic structure and the expected chemical stability of untreated carbon fibres. The next significant component, attributed to C-OH/C-O-C groups at 286.6 eV, comprises 29.7%. These oxygen-containing groups are likely associated with the sizing layer applied to virgin fibres and their surface treatment. Minor contributions from C=O groups (0.9%) and COOR groups (2.2%) are also observed.

In the case of Pl_rCF (Figure 9), the C-C/C-H component remains the predominant signal at 285.0 eV at 69.7%. However, there is a notable reduction in C-OH/C-O-C groups, which decreased to 15.1%. Concurrently, the C=O and COOR peaks increase to 5.7% and 7.5%, respectively, reflecting the introduction of oxygen-containing functional groups due to the plasma treatment. These groups are associated with surface oxidation processes and likely contribute to improved surface energy and potential bonding capabilities. Additionally, a small carbonate peak appears at 291.2 eV (2.0%).

Sized_Pl_rCF (Figure 10) shows significant changes in surface chemistry due to the application of sizing. The C-C/C-H component decreases to 48.8% while the C-OH/C-O-C group increases to 30.5%. This rise indicates that the sizing process introduces or enhances hydroxyl and ether functionalities, critical for fibre-matrix adhesion. The C=O and COOR peaks also show substantial increases, reaching 8.7% and 9.3%, respectively. These increases highlight the introduction of functional groups during sizing, which further enhance the fibre’s interfacial bonding. The small carbonate peak at 290.5 eV (2.6%) remains evident.

In addition to the C1s spectra, the high-resolution O1s peaks provide further information into the oxygen-containing functional groups present on the surface of the carbon fibres. Table 10 shows the deconvoluted O1s peaks for each of the fibre types, highlighting the specific oxygen functionalities.

For the Ref_CF, the O1s spectrum is dominated by the C-OH group at 533.1 eV, accounting for 91.8% of the total oxygen content. This indicates a high concentration of hydroxyl functionalities, likely resulting from sizing agents and surface treatments. Additionally, smaller peaks are observed, with O=C at 531.3 eV contributing 4.1% and C-O/ester groups at 534.3 eV also contributing 4.1% of the oxygen content. In the case of Pl_rCF, significant changes in oxygen chemistry are observed. The C-OH content decreases to 36.7%, and new oxygen-containing species emerge. Concurrently, the O=C/O=C-O-R component at 531.9 eV increases substantially to 35.5%, indicating surface modifications introduced during the recycling process. Additionally, new peaks appear, including O=C-O-R/O-N/C-O at 534.0 eV, accounting for 17.5% of the oxygen content. These functional groups are likely formed as a result of plasma treatment in the HNO_3_ solution, which introduces nitrogen-oxygen species such as nitro (–NO_2_) or nitroso (–NO) groups through oxidative reactions. A small peak at 535.8 eV (1.5%) is associated with adsorbed water, indicating potential interactions with the aqueous medium used during the recycling process. For the Sized_Pl_rCF, the O1s spectrum reveals further modifications. The C-OH content increases to 42.0%. The sizing process likely introduced or retained hydroxyl groups on the fibre surface, contributing to improved potential for fibre-matrix adhesion. The O=C/(O=C-O-R) component at 532.0 eV decreases to 7.0%, while the O=C-O-R/O-N/C-O group at 534.0 eV increases to 31.7%, indicating further oxidation/surface modification due to the sizing. The water-related peak at 535.7 eV becomes more pronounced, contributing 13.5% of the oxygen content. This is likely due to the aqueous nature of the sizing solution, which may have left residual water molecules or OH• and •HO_2_ radicals that are produced during the plasma in bubbles processing and are chemically attached to the fibre surface. The presence of these species suggests that the sizing process not only chemically modifies the surface but also introduces additional functional groups that can enhance fibre-matrix adhesion by increasing the availability of reactive sites for covalent bonding. The overall increase in oxygen-containing groups demonstrates that the sizing treatment significantly alters the surface chemistry of Pl_rCFs. This transformation is expected to improve the interfacial adhesion between the fibre and matrix, which is crucial for the mechanical performance of composite materials.

#### 3.3.4. Mechanical Tests

To further evaluate the effectiveness of the immersion of rCFs in polymeric coatings, fibres were also tested over their mechanical strength, and the results were compared to those of Ref_CFs. The results of the measurements are presented in Table 11.

Both recycling processes resulted in similar outcomes, showing approximately a 20% reduction in tensile strength compared to Ref_CF. This reduction in strength is likely due to the non-uniform shape of Pl_rCFs and Ch_rCFs and material degradation. Additional factors that may impact the quality of Pl_rCFs and Ch_rCFs include irregularities in fibre morphology, disordered fibre distribution within the composite, and poor bonding between the fibre and the matrix. However, Sized_Pl_rCF and Sized_Ch_rCF yarns exhibited only a 10% decrease in tensile strength relative to Ref_CF. This confirms that applying polymeric coatings to Pl_rCFs and Ch_rCFs enhances their mechanical properties by mitigating the negative effects mentioned earlier and making the fibres more consistent and durable. Tensile strength largely depends on the Young’s modulus of the material. Although sizing does not alter the intrinsic Young’s modulus of the fibre, it can improve fibre-matrix adhesion, which is another key factor contributing to a material’s tensile strength [26,27]. The slight increase in tensile strength obtained by this further treatment could potentially be attributed to this enhanced adhesion.

As for the intrinsic Young’s modulus of CFs, it is generally expected to remain unaffected by the recycling and sizing processes. However, the results indicated a lower measured modulus for the fibres from both recycling methods (chemically assisted solvolysis and plasma-enhanced solvolysis). A possible explanation is that the separation of the fibre from the matrix during recycling can introduce surface roughness on the CFs (however, this is not confirmed from SEM analysis). Additionally, fibre misalignment during sample preparation for tensile testing may contribute to the lower modulus values, as fibre orientation directly influences load transfer efficiency, leading to reduced stiffness measurements. For Sized_Pl_rCFs and Sized_Ch_rCFs, there was a partial restoration of the modulus, although it remained lower than that of the Ref_CF. This improvement is likely to be due to the restoration of surface smoothness and the protection that sizing provides against mishandling and other external factors. These enhancements enable more effective load transfer and stress distribution during tensile testing.

Tensile testing was also conducted on rCF monofilaments according to the process mentioned in Section 2.5. The respective results are presented in Table 12. To provide further context for the tensile testing results, a comparison table (Table 13) is presented. It includes tensile strength data for virgin carbon fibres (vCFs) and rCFs obtained using various recycling methods. Comparison with previous studies shows a range of tensile strength reductions, influenced by the recycling method used. For instance, pyrolysis often leads to a slight reduction due to the thermal degradation of the polymer matrix, which can leave behind char residues that interfere with fibre-matrix bonding. Microwave pyrolysis, despite being a rapid process, can introduce localised overheating, causing surface defects and impacting tensile strength. In super/subcritical water-acetone solvolysis, high-pressure and high-temperature conditions can result in fibre oxidation or minor structural disruptions. Similarly, chemical solvolysis using acids or bases may oxidise the surface or leave behind residual functional groups that alter fibre properties. For all these methods, post-recycling treatments, such as surface cleaning or re-sizing, are crucial to restore or improve fibre-matrix adhesion and tensile strength.

The tensile testing results for single CFs, presented in Table 12, demonstrate consistent trends with the findings from the bundle tensile tests shown in Table 11. Both tests reveal that Pl_rCFs and Ch_rCFs exhibit a reduction in tensile strength compared to Ref_CF, with decreases of approximately 15.6% and 12.5%, respectively, for single fibres. This is in line with the ~19% reduction observed in bundle tests. For both recycling methods, the tensile strength of the sized fibres shows significant improvement. The Sized_Ch_rCFs exhibited an 8.6% decrease, and the Sized_Pl_rCFs showed an 8.3% decrease in strength compared to Ref_CFs, further confirming the effectiveness of polymeric sizing. This aligns with the bundle tensile tests, where the reduction was approximately 9%. The application of polymeric coatings mitigates the negative effects of the recycling process, enhancing fibre-matrix adhesion and providing better load transfer during mechanical testing. While tensile strength depends largely on the intrinsic Young’s modulus of the material, these improvements likely depend on the well-aligned fibres and better adhesion after sizing.

## 4. Conclusions

The aim of this study was to investigate the impact of two chemical recycling methods—chemically-assisted and plasma-assisted solvolysis—on the morphology and properties of CFs derived from end-of-life CFRP automotive parts, as well as the effect of sizing application on these recycled fibres. SEM analysis was used to examine fibre surface morphology, revealing that chemically assisted solvolysis left some resin residues, while plasma-assisted solvolysis was more effective at removing them. Sized fibres displayed a smooth surface with a uniform coating distribution. The structural integrity of both the retrieved and sized fibres was assessed through TGA and Raman spectroscopy. TGA measurements confirmed that all fibre types maintained their structural integrity, showing no mass loss up to 900 °C. Comparing Raman spectra, it became evident that plasma-assisted solvolysis more effectively retains fibre integrity by limiting the introduction of disorder and reducing residual resin, whereas traditional solvolysis showed the presence of functional groups such as C=O and OH, suggesting the residual presence of epoxy resin together with an advance oxidation of the surface.

Additionally, the study compared different methods for analysing optical microscopy images, each with specific strengths and limitations. Olympus software emerged as the fastest method, enabling immediate analysis post-image capture, but it struggled with dense or overlapping fibres. ImageJ, though more time-consuming, proved to be very consistent, offering a step-by-step process that ensured high accuracy. Its watershed algorithm was particularly effective in segmenting overlapping or adjacent fibres, enhancing precision. Finally, a Python script was developed, allowing for automated and efficient analysis with faster processing times.

To determine the effect of sizing on the properties of recycled CFs, XPS analysis was conducted. The results showed an increase in functional groups on the fibre surfaces, which improved their mechanical properties. Tensile tests confirmed that both sized and unsized recycled fibres exhibited a reduction in tensile strength of approximately 10% compared to reference fibres, as expected with recycled materials. However, the application of sizing significantly enhanced the tensile strength of the recycled fibres, mitigating the negative effects of recycling, such as fibre damage, misalignment, and reduced matrix adhesion.

While plasma-assisted solvolysis proved to be more effective at removing resin, both chemical recycling methods successfully reclaimed the fibres. The application of sizing further enhanced the performance of the recycled fibres by improving their mechanical strength and mitigating the drawbacks of the recycling process, demonstrating that sizing plays a crucial role in restoring the functionality of recycled carbon fibres.

## 5. Outlook and Future Directions

This study puts emphasis on the critical role of chemical recycling and surface modification techniques in advancing the sustainability and functionality of carbon fibre-reinforced composites. However, to fully integrate these processes into industrial applications, further research and development are required.

One promising direction for future research involves optimising chemical recycling methods to minimise structural disruptions during fibre recovery. By refining solvent systems and process parameters, it may be possible to reclaim fibres with enhanced structural and surface properties, further reducing the gap between recycled and virgin fibres. Moreover, with optimised processes, it could be possible to achieve complete resin removal during recycling, eliminating the need for additional post-treatment steps to address residual resin. This approach would enhance the overall efficiency of the recycling process and ensure that the recovered fibres are immediately suitable for surface modification processes, such as sizing, which are essential for restoring fibre-matrix adhesion and mechanical performance.

Advancements in surface modification methods also offer significant potential. The incorporation of nanomaterials, such as carbon nanotubes or nanoparticles, into sizing formulations could introduce multifunctional properties with enhanced mechanical performance, improved thermal stability, or electrical conductivity. Such innovations would broaden the application scope of recycled carbon fibres, enabling their use in more demanding composite applications.

Automation and real-time monitoring of recycling processes represent another critical area for development. The Python-based image analysis introduced in this study demonstrates the potential for quantitative evaluation of fibre quality, but further refinement is needed to enable real-time integration into recycling workflows. Automated feedback systems could provide valuable data to adjust process parameters dynamically, ensuring consistent and high-quality fibre recovery.

Finally, scaling up these recycling processes from laboratory to industrial scale will be key to addressing the growing demand for sustainable composite materials. This will require collaboration between industry stakeholders, researchers, and policymakers to develop efficient, cost-effective, and environmentally friendly recycling solutions. Establishing standardised evaluation methods for recycled fibres and demonstrating their reliability in real-world applications will be essential to gaining industry acceptance and fostering broader adoption.

## Figures and Tables

**Figure 1 polymers-17-00033-f001:**
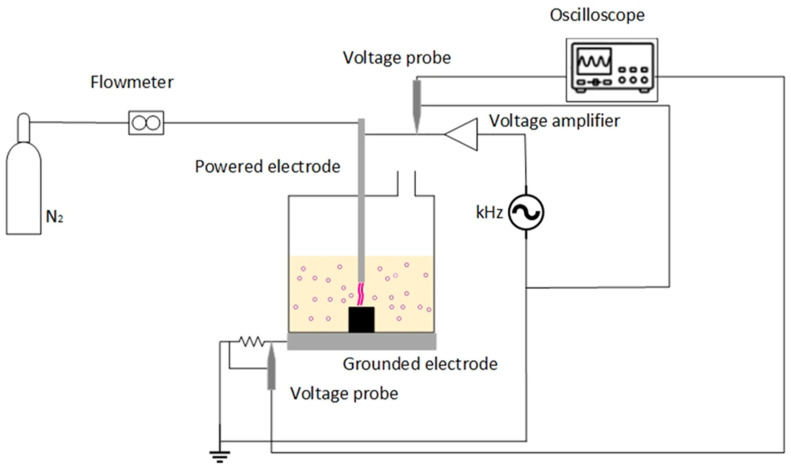
Plasma reactor set-up.

**Figure 2 polymers-17-00033-f002:**
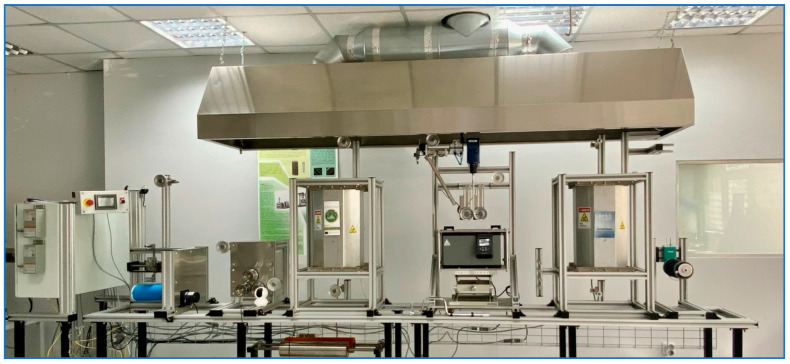
Fibre sizing line.

**Figure 3 polymers-17-00033-f003:**
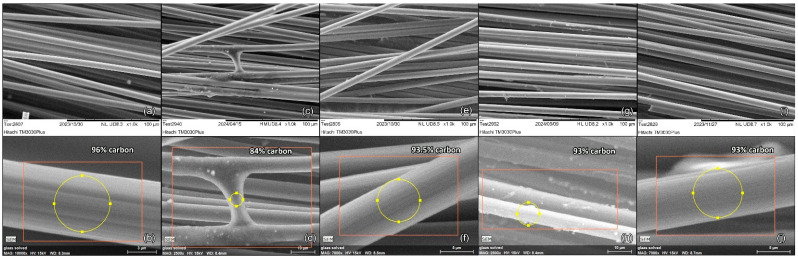
SEM images of (**a**) Ref_CF and (**b**) EDS spot, (**c**) Ch_rCF and (**d**) EDS spot, (**e**) Pl_rCF and (**f**) EDS spot, (**g**) Sized_Ch_rCF and (**h**) EDS spot, (**i**) Sized_Pl_rCF and (**j**) EDS spot.

**Figure 4 polymers-17-00033-f004:**
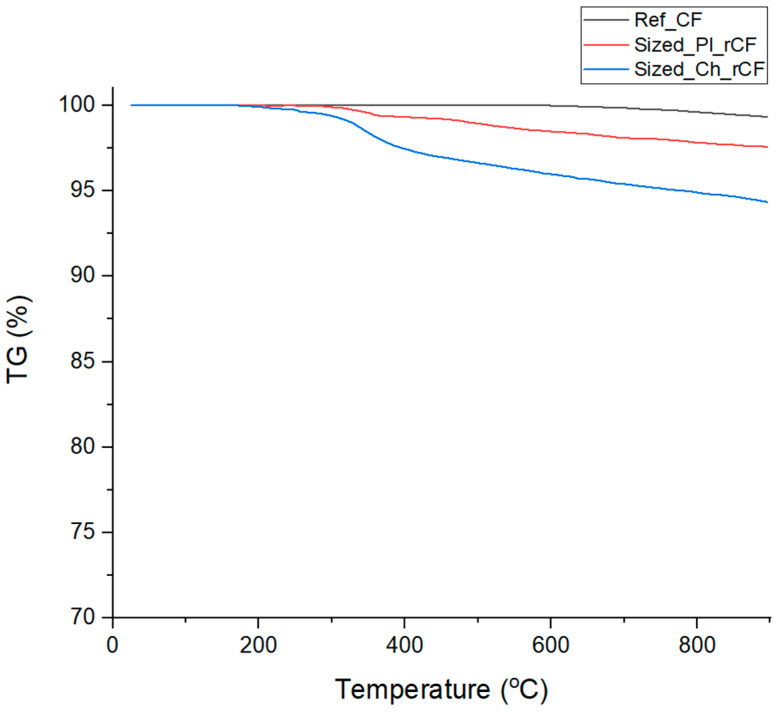
TGA results of Ref_CFs, Sized_Pl_rCFs and Sized_Ch_rCFs.

**Figure 5 polymers-17-00033-f005:**
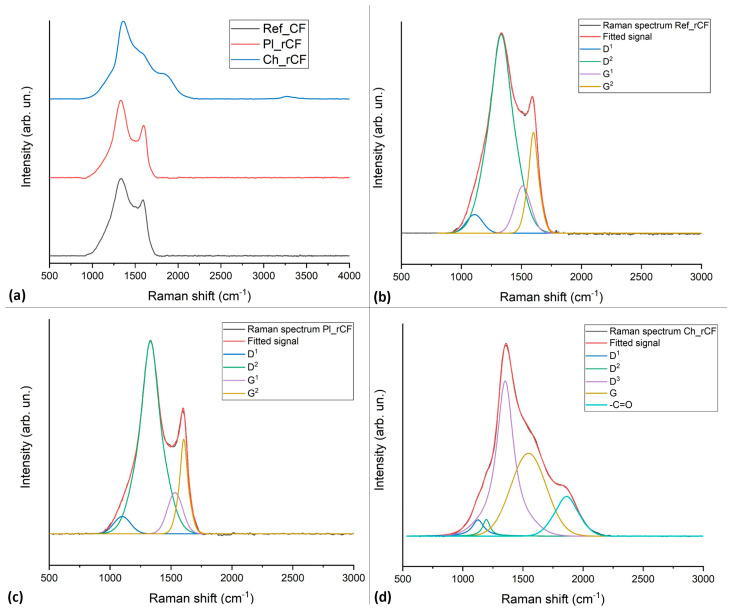
Raman spectra of (**a**) Ref_CFs, Pl_rCFs, and Ch_rCFs and deconvoluted spectra illustrating fitting signals for (**b**) Ref_CFs, (**c**) Pl_rCFs and (**d**) Ch_rCF.

**Figure 6 polymers-17-00033-f006:**
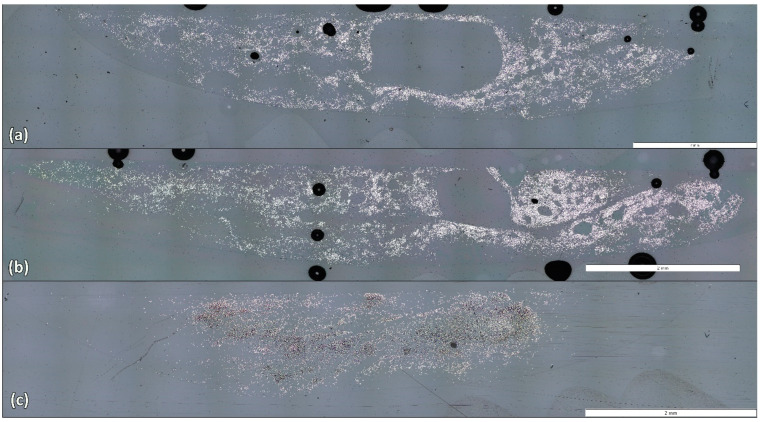
(**a**) Ref_CF, (**b**) Pl_rCF, and (**c**) Ch_rCF.

**Figure 7 polymers-17-00033-f007:**
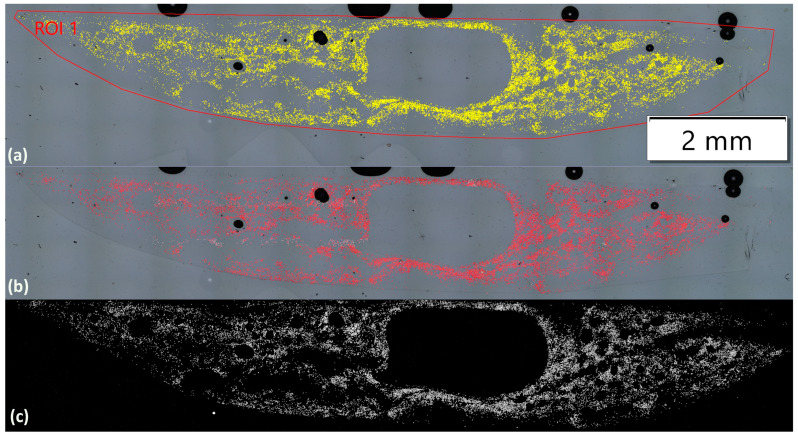
(**a**) Ref_CF from Olympus software, (**b**) Ref_CF from Python, and (**c**) Ref_CF from ImageJ software.

**Figure 8 polymers-17-00033-f008:**
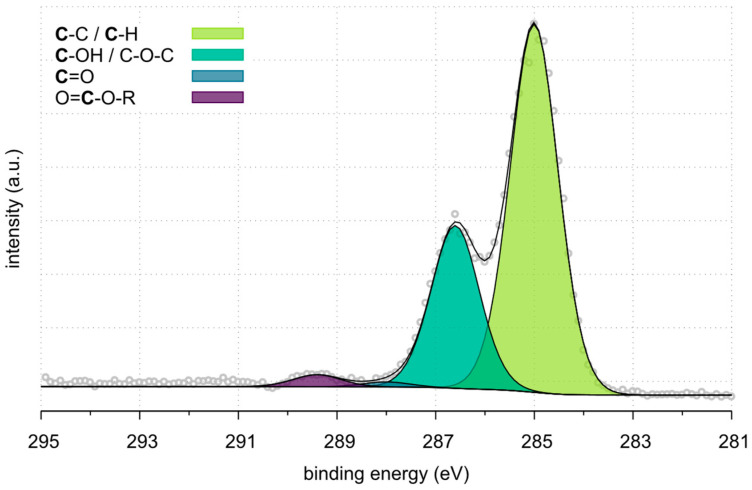
HR spectra of Ref_CF.

**Figure 9 polymers-17-00033-f009:**
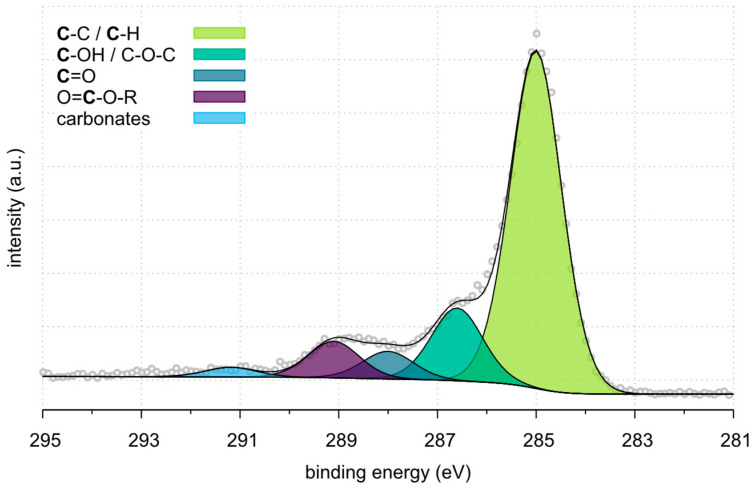
HR spectra of Pl_rCF.

**Figure 10 polymers-17-00033-f010:**
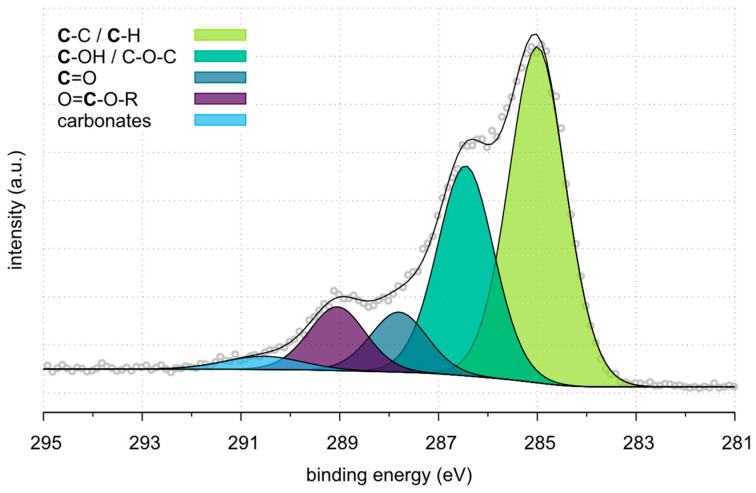
HR spectra of Sized_Pl_rCF.

**Table 1 polymers-17-00033-t001:** Datasheet of Tenax^®^-E STS40.

Tenax^®^-E STS40 E23 24K 1600tex	
Tensile strength (MPa)	4300
Tensile modulus (GPa)	250
Elongation at break (%)	1.7
Density (g/cm^3^)	1.78
Filament diameter (μm)	7

**Table 2 polymers-17-00033-t002:** Datasheet of Hydrosize^®^ HP2-06.

Physical Properties	
pH	6.5–8.5
Emulsifier Charge	Amine-dispersed
Percent Non-Volatile (%)	24.5–26.5
Recommended pH Range	6.5–8.5
Brookfield Viscosity Range (cps)	<2000
Appearance	White emulsion

**Table 3 polymers-17-00033-t003:** Datasheet of SR1710 injection/SD8822.

Technical Properties	
Modulus of elasticity (GPa)	3.65
Elongation at break (%)	2.2
Flexural Strength (MPa)	115
Charpy impact strength (kJ/m^2)^	17
Glass Transition Temperature (°C)	53
Tensile Strength (MPa)	67

**Table 4 polymers-17-00033-t004:** Overview of the five sample categories studied in this manuscript and their corresponding references for clarity.

Sample Type	Code Name
Reference carbon fibre (untreated)	Ref_CF
Plasma solvolyzed carbon fibre	Pl_rCF
Chemically solvolyzed carbon fibre	Ch_rCF
Sized plasma solvolyzed carbon fibre	Sized_Pl_rCF
Sized chemically solvolyzed carbon fibre	Sized_Ch_rCF

**Table 5 polymers-17-00033-t005:** Grinding parameters.

Paper Type	Suspension Type	Force (N)	Turns per Minute	Time (min)
SiC 220	Water	5	300	0:30
SiC 220	Water	10	300	0:30
SiC 500	Water	10	300	1:00
SiC 1200	Water	10	300	1:00
SiC 2200	Water	10	300	1:00
SiC 4000	Water	10	300	1:00

**Table 6 polymers-17-00033-t006:** Polishing parameters.

Paper Type	Suspension Type	Force (N)	Turns per Minute	Time (min)
MD-Largo	DiaPro Largo 3 μm	10	150	5:00
MD-Dur	DiaPro Nap 1 μm	10	150	5:00

**Table 7 polymers-17-00033-t007:** Filament count master table.

	Ref_CF		Pl_rCF		Ch_rCF	
Method	Filament Count	Error (%)	Filament Count	Error (%)	Filament Count	Error (%)
Olympus software	14,856 ± 1294	8.7	14,749 ± 558	3.8	13,926 ± 1556	11.2
ImageJ	23,700 ± 1391	5.9	22,501 ± 287	0.3	21,930 ± 1118	5.1
Python	23,725 ± 820	3.5	22,247 ± 303	1.4	22,527 ± 873	3.9

**Table 8 polymers-17-00033-t008:** Elemental composition of CFs.

%	Ref_CF	Pl_rCF	Sized_Pl_rCF
C 1s	83.4 ± 0.04	75.7 ± 0.1	79 ± 0.05
O 1s	14.9 ± 0.03	18.5 ± 0.04	19.3 ± 0.04
N 1s	-	4 ± 0.1	0.6 ± 0.03
Si 2p	0.9 ± 0.02	0.6 ± 0.02	0.6 ± 0.02
Cl 2p	0.4 ± 0.02	0.2 ± 0.01	-
Ca 2p	-	0.9 ± 0.02	0.5 ± 0.01

**Table 9 polymers-17-00033-t009:** Binding energy and percentage of carbon bonds.

	Ref_CF		Pl_rCF		Sized_Pl_rCF	
Name	Position	% At Conc	Position	% At Conc	Position	% At Conc
C-C/C-H	285.0	67.2	285	69.7	285	48.8
C-OH/C-O-C	286.6	29.7	286.6	15.1	286.5	30.5
C=O	288	0.9	288	5.7	287.8	8.7
COOR	289.4	2.2	289.1	7.5	289.1	9.3
Carbonates			291.2	2	290.5	2.6

**Table 10 polymers-17-00033-t010:** Binding energy and percentage of oxygen bonds.

	Ref_CF		Pl_rCF		Sized_Pl_rCF	
Name	Position	%At Conc	Position	%At Conc	Position	%At Conc
O-Ca			531.1	8.8	531.1	5.1
O=C/(O=C-O-R)	531.3	4.1	531.9	35.5	532	7
C-O/O=C-O-R	534.3	4.1				
C-OH	533.1	91.8	533	36.7	532.8	42.7
O=C-O-R-/O-N/C-O			534	17.5	534	31.7
H_2_O			535.8	1.5	535.7	13.5

**Table 11 polymers-17-00033-t011:** Results of tensile testing on CFs.

Specimen	Tensile Strength (GPa)	% Difference to Ref_CF	Modulus (GPa)	% Difference to Ref_CF
Ref_CF	2.7 ± 0.3	N/A	180 ± 45	N/A
Ch_rCF	2.2 ± 0.2	−18	140 ± 12	−22
Sized_Ch_rCF	2.4 ± 0.3	−11	169 ± 7	−7
Pl_rCF	2.1 ± 0.3	−22	151 ± 24	−17
Sized_Pl_rCF	2.4 ± 0.4	−11	163 ± 9	−10

**Table 12 polymers-17-00033-t012:** Results of tensile testing on single CFs.

Specimen	Tensile Strength (GPa)	% Difference to Ref_CF
Ref_CF	3.60 ± 0.38	N/A
Ch_rCF	3.04 ± 0.48	−15.6
Sized_Ch_rCF	3.29 ± 0.21	−8.6
Pl_rCF	3.15 ± 0.38	−12.5
Sized_Pl_rCF	3.30 ± 0.28	−8.3

**Table 13 polymers-17-00033-t013:** Comparison of tensile strength for vCFs and rCFs obtained using various recycling methods.

Citation	Method	vCF (GPa)	rCF (GPa)	% Difference
G. Jiang & S. Pickering [28]	Pyrolysis	7 ± 2.4	5.9–6.8 ± 1.4–2.3	2.9–15.7
Y. Ren et al. [29]	Microwave Pyrolysis	3.1 ± 0.22	2.2–3 ± 0.32–0.53	3.2–29
C. Vogiantzi & K. Tserpes [30]	Sub/Super critical water-acetone solvolysis	2.7 ± 1.03	1.8–2.5 ± 0.77–1.29	6.4–31.2
X. Zhang et al. [31]	Chemical solvolysis	3.2 ± 1.18	2.5–3 ± 0.99–1.35	6.3–21.9
C. Chaabani et al. [32]	Sub/Super critical water solvolysis	4.9 ± 0.5	4.1–4.5	8.2–16.3

## Data Availability

Data will be available on the Zenodo platform upon request with the following DOI: https://doi.org/10.5281/zenodo.14535117 (accessed on 19 December 2024).

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
