# Peer review of "Performance Restoration of Chemically Recycled Carbon Fibres Through Surface Modification with Sizing"

_polymers, 2024, doi:10.3390/polym17010033_

Round 1
Reviewer 1 Report
Comments and Suggestions for Authors
This work deals with very important topic - recycling of the CRF-loaded polymers. Indeed, all processes are complex and multi-step methods. The authors showed very interesting results, but some interpretations are not correct:
1) XPS results: both C1s and O1s spectra fits are made with severe mistakes: the C=C sp2 contribution has asymmetric shape, although it not proven with alternative data that the surface has graphitic carbon - although it is easy to understand by analysis the Auger line (C KLL, from the same XPS spectrum). Secondly, the position of the C-H peak is shifting from sample to sample! This will lead to critical changes in the intensity of C=C and CH peaks and, as a result, misleading results. The positions of these two peaks must be fixed! Finally the C==C is covering the full peak of the CH and, therefore, all conclusions made form this part of fitting are not correct. Please make fitting very carefully! Use only first decimals and fix both FWHM and positions for all peaks in the spectra! (It is too big variations from 1.1 eV to 1.5 eV FWHM from sample to sample. Finally, the positions of the O1s peaks are wrong. Look to the Beamson and Briggs table. N-O was never 532.1 eV! Check other peaks as well!!! Please improve also the quality of XPS figures.
2) There are problems with the crosslinks, everywhere you see
Error! Reference source not found.
3) The methodologies of the recycling should be also understood from the position of the cost and energy efficiency. At least very high level comparison is important!
4) Please provide outlook for this are of research.
Author Response
REVIEWER 1:
1) XPS results: both C1s and O1s spectra fits are made with severe mistakes: the C=C sp2 contribution has asymmetric shape, although it not proven with alternative data that the surface has graphitic carbon - although it is easy to understand by analysis the Auger line (C KLL, from the same XPS spectrum). Secondly, the position of the C-H peak is shifting from sample to sample! This will lead to critical changes in the intensity of C=C and CH peaks and, as a result, misleading results. The positions of these two peaks must be fixed! Finally the C==C is covering the full peak of the CH and, therefore, all conclusions made form this part of fitting are not correct. Please make fitting very carefully! Use only first decimals and fix both FWHM and positions for all peaks in the spectra! (It is too big variations from 1.1 eV to 1.5 eV FWHM from sample to sample. Finally, the positions of the O1s peaks are wrong. Look to the Beamson and Briggs table. N-O was never 532.1 eV! Check other peaks as well!!! Please improve also the quality of XPS figures.
RESPONSE: Thank you very much for your detailed comments on the XPS analysis. The operator made incorrect assumptions about the tested material, hence the asymmetric C=Csp2 peak. Thanks to your comment, this error was noticed and a new fitting was performed based on better data and suggested publications. New, better quality figures were provided and care was taken to only first decimals where shown. Once again, thank you for your vigilance, as it helped us improve the quality of the analysis and the conclusions drawn.
2) There are problems with the crosslinks, everywhere you see
Error! Reference source not found.
RESPONSE: Thank you for the comment. We have carefully reviewed all the crosslinks in the text and corrected the errors related to the "Error! Reference source not found." issue. The correct cross-references for tables and figures have now been added throughout the manuscript.
3) The methodologies of the recycling should be also understood from the position of the cost and energy efficiency. At least very high level comparison is important!
RESPONSE: thank you for this comment. That is correct, the discussion and highlighting the energy and cost efficiency of the proposed methodologies are absolutely crucial. We have added a discussion under the "Methods" section addressing the energy and cost efficiency of the two recycling methods. Both methods are currently at low technology readiness levels (TRLs 3–5), and their laboratory-scale energy profiles provide insights for future optimization efforts as they move toward industrial adoption.
4) Please provide outlook for this area of research.
RESPONSE: Thank you for your suggestion. In response, we have added a Outlook and Future Directions section within the manuscript. This section highlights key areas for future work based on the experience we gained from implementing the work described in the manuscript.
Reviewer 2 Report
Comments and Suggestions for Authors
Semitekolos et al., describe the effect of chemical and plasma enhanced solvolysis based recycling carbon fibres (CFs) recovered from end-of-life automotive parts. Additionally, the impact of fibre sizing is studied to improve the functional surface of the recycled carbon fibres (rCFs). SEM, TGA, and Raman spectroscopy were used to determine the surface morphology and integrity of the rCFs. Filamental loss during the recycling process quantified via automated analysis of the optical microscopy images using Software and Python algorithms is exciting. The application of sizing helped to recover 50% of the tensile strength lost due to the solvolysis process. Surface charge was analyzed by X-ray Photoelectron Spectroscopy (XPS) which further endorsed the presence of abundant oxygen-containing groups on the fibre surface, which facilitates fibre-matrix hold in the milieu. Overall, the work describes an effective methodology to recycle Carbon Fibres which are in increasing demand for sustainable raw materials in industries. However, several concerns need to be addressed before considering publishing.
1. The introduction does not highlight or indicate the innovation in this work. Hence the results appear to be like typical increment to the field. A thorough rework of the introduction is essential. The authors fail to illustrate the novelty of the approach used here, especially in the objective section (Page 3 lines 103-111). Specially they mention two chemically assisted recycling methods used here but fail to discuss adequate background to the same. Encompassing late case studies of the same approaches can improve the understanding and impact of the same. On page 2 line 78 the authors state that ‘’Three of the most studied chemical recycling techniques include alcoholysis, oxidation, and the use of sub- or super-critical fluids. Alcoholysis typically operates under mild conditions (100-200°C and 1-5 MPa) but is hindered by the high boiling points of the solvents, which can lead to incomplete decomposition and extended processing times. Wet oxidation uses acidic solutions like nitric acid or hydrogen peroxide at lower temperatures and atmospheric pressure [13]. The use of sub- or super-critical solvents (e.g., water, organic solvents) offers faster decomposition and higher efficiency [14].’’ However, it lacks a clear literature analysis that outlines similar methods explored, the subsequent merits of the existing methodologies and the gaps left for exploration. Especially the significance of the chemical and plasma method used here needs to be discussed considering previous similar reports by the same group (see Ref 16). Finally, my understanding is that the fundamental goal of the study is sizing on recycled CFs. Clarifying this will substantially boost the impact and coherence of the work to readers of the same background. Please remove references (Ref 16) from the objective section as the results are new and instead discuss the same in the case study.
22. It will be worth discussing the chemistry of the sizing process used here in a general scheme. (The hydrasize sizing agent used here is a phenoxy dispersion in an amine emulsifier). How is it going to attach to the rCF surface?
3. The microscopy discussion section using SEM images is not clear in terms of the sizing effect on the chemical solvolysis RCFs. If the surface is going to remain the same even after sizing, then what is the effect of sizing? The main merit is the use of plasma recycling, and the additional sizing is unnecessary for binding to resin in composite formation. This is clear in some of the statements. Please see this statement on Page 9 line 336 ‘’This oxygen presence can be attributed to plasma treatment, which not only assists in removing residual polymer matrix but also introduces oxygen-containing functional groups, such as carboxyl (-COOH) groups, onto the fibre surface. These functional groups enhance the surface energy of the fibres, promoting stronger adhesion to the resin matrix and improving the overall bonding performance in composite applications. ‘’ Additionally, Figures 3b-j denote a zoomed area of the SEM images shown in the upper panel, not EDS data. Please include the EDS graph showing Carbon and the Oxygen percentage of the area in the supporting file.
4. The TGA data indicates that sizing slightly affects the temperature stability of the RCFs, especially in the case of the chemical solvolysis method. Is it because of the higher residue remnant in the sample? The authors should also consider unreacted organic polymer coating (sizing agent) towards this behavior. Comparison with only Recycled CFs (Chemical &Plasma Induced) is missing in the data. The sample labelling in the graph is not clear. Please indicate in the legend of Fig 4 as TGA results of Ref_CFs (Green graph), Sized_Pl_rCFs (Red graph) and Sized_Ch_rCFs (Blue graph).
5. Figure 5 shows the normal and deconvoluted Raman spectra of only the reference and Recycled CFs. Additionally, the effect of functionalization via sizing can be understood by providing Raman or FTIR data of sized RCFs. A discussion regarding quantitation based on a deconvoluted percentage population of different functionalities in the Ref, Recycled and sized samples may be helpful to readers to gain a deeper understanding of the functionalization.
6. Additionally, a comparison table with attributes like Tensile Strength obtained for CFs via this approach with other previously reported methods can be interesting to be shown in a table which I believe will increase the impact of the work.
7. Minor comment
There are several areas in the manuscript where I saw an error message like Error! Reference source not found. I believe this may be because of some error due to referencing software issues. Please rectify this
Author Response
REVIEWER 2:
- The introduction does not highlight or indicate the innovation in this work. Hence the results appear to be like typical increment to the field. A thorough rework of the introduction is essential. The authors fail to illustrate the novelty of the approach used here, especially in the objective section (Page 3 lines 103-111). Specially they mention two chemically assisted recycling methods used here but fail to discuss adequate background to the same. Encompassing late case studies of the same approaches can improve the understanding and impact of the same. On page 2 line 78 the authors state that ‘’Three of the most studied chemical recycling techniques include alcoholysis, oxidation, and the use of sub- or super-critical fluids. Alcoholysis typically operates under mild conditions (100-200°C and 1-5 MPa) but is hindered by the high boiling points of the solvents, which can lead to incomplete decomposition and extended processing times. Wet oxidation uses acidic solutions like nitric acid or hydrogen peroxide at lower temperatures and atmospheric pressure [13]. The use of sub- or super-critical solvents (e.g., water, organic solvents) offers faster decomposition and higher efficiency [14].’’ However, it lacks a clear literature analysis that outlines similar methods explored, the subsequent merits of the existing methodologies and the gaps left for exploration. Especially the significance of the chemical and plasma method used here needs to be discussed considering previous similar reports by the same group (see Ref 16). Finally, my understanding is that the fundamental goal of the study is sizing on recycled CFs. Clarifying this will substantially boost the impact and coherence of the work to readers of the same background. Please remove references (Ref 16) from the objective section as the results are new and instead discuss the same in the case study.
RESPONSE: We appreciate the reviewer’s valuable feedback. To address this comment, we have thoroughly revised the Introduction section to better highlight the innovation and novelty of the work. We provided a more detailed background on chemical recycling techniques. We restructured the objective section to highlight the role of sizing and we tried to focus more on the novelties that this work is bringing on the readers of Polymers journal.
- It will be worth discussing the chemistry of the sizing process used here in a general scheme. (The hydrasize sizing agent used here is a phenoxy dispersion in an amine emulsifier). How is it going to attach to the rCF surface?
RESPONSE: Please check the file since for this answer we have also included figure in the response that are not visible here.
The hydrasize sizing agent used here is a phenoxy dispersion in an amine emulsifier. Phenoxy resins in a mixture with amines will generate a hydroxyl group in the polymeric chain through the interaction of the phenoxy ester or ether with the amine group. The hydroxyl group will promote the covalent bonding of the sizing polymer to the rCF surface through either dehydration or esterification reactions with -OH or -COOH groups that already exist in rCF surface. In addition, the hydroxyl group will enhance h-bonding interactions between the sizing polymer and rCFs surface leading thus to a stronger attachment.
A possible sketch of the sizing polymer attachment to the rCF surface is presented below for further clarification. We don’t wish to add the sketch in the paper as we don’t know the exact chemical composition of Hydrosize® HP2-06 but we have added the above-mentioned sentence in line 228 to give a general idea of the sizing chemical process.
Step 1 – Hydroxyl group formation
Step 2 (a) covalent bonding to rCF surface
Step 2 (b) H- bonding to rCF surface
- The microscopy discussion section using SEM images is not clear in terms of the sizing effect on the chemical solvolysis RCFs. If the surface is going to remain the same even after sizing, then what is the effect of sizing? The main merit is the use of plasma recycling, and the additional sizing is unnecessary for binding to resin in composite formation. This is clear in some of the statements. Please see this statement on Page 9 line 336 ‘’This oxygen presence can be attributed to plasma treatment, which not only assists in removing residual polymer matrix but also introduces oxygen-containing functional groups, such as carboxyl (-COOH) groups, onto the fibre surface. These functional groups enhance the surface energy of the fibres, promoting stronger adhesion to the resin matrix and improving the overall bonding performance in composite applications. ‘’ Additionally, Figures 3b-j denote a zoomed area of the SEM images shown in the upper panel, not EDS data. Please include the EDS graph showing Carbon and the Oxygen percentage of the area in the supporting file.
RESPONSE: Thank you for your comment. We have carefully addressed each point as follows:
Clarification on the Sizing Effect: We agree that the original discussion required further clarification regarding the role of sizing in chemically solvolyzed rCFs. While the plasma treatment introduces oxygen-containing functional groups, such as carboxyl (-COOH) groups, which enhance fiber-matrix adhesion, these groups alone may not provide sufficient durability and mechanical stability during composite manufacturing. Sizing plays a complementary role by forming a protective polymeric layer on the fiber surface, which not only further enhances the interfacial bonding but also protects the fibers during handling and processing. We’ve included this justification inside the manuscript.
Additional EDS Data: We have now included the EDS graphs showing the carbon and oxygen percentages in the supplementary materials, as requested.
Figures 3b-j Clarification: We have revised the figure legends to mention that this is the spot were EDS analysis have been performed.
- The TGA data indicates that sizing slightly affects the temperature stability of the RCFs, especially in the case of the chemical solvolysis method. Is it because of the higher residue remnant in the sample? The authors should also consider unreacted organic polymer coating (sizing agent) towards this behavior. Comparison with only Recycled CFs (Chemical & Plasma Induced) is missing in the data. The sample labelling in the graph is not clear. Please indicate in the legend of Fig 4 as TGA results of Ref_CFs (Green graph), Sized_Pl_rCFs (Red graph) and Sized_Ch_rCFs (Blue graph).
RESPONSE: Thank you for your comment. Through this graph, it is indicated that after chemically assisted solvolysis, a larger quantity of resin residue can be observed on those fibres, which results to higher mass loss after the performance of TGA. From our point of view, it was crucial to compare the reference fibres, to which commercial sizing had been applied, with the recycled ones, after the deposition of the sizing material. The total graph has been redrawn, in order to be clearer.
- Figure 5 shows the normal and deconvoluted Raman spectra of only the reference and Recycled CFs. Additionally, the effect of functionalization via sizing can be understood by providing Raman or FTIR data of sized RCFs. A discussion regarding quantitation based on a deconvoluted percentage population of different functionalities in the Ref, Recycled and sized samples may be helpful to readers to gain a deeper understanding of the functionalization.
RESPONSE: Thank you for the comment. We understand the importance of such analysis; however, it is important to highlight some intrinsic limitations of our materials and processes that prevent us from fulfilling the request. The reference carbon fiber (Ref) is a commercial product that contains sizing. This means that its Raman spectrum is not only representative of the carbon structure but also includes contributions from the sizing material. Additionally, after the fiber is used in the fabrication of a composite with epoxy resin matrix and subsequently recycled, it is not possible to determine with certainty the exact amount of residual sizing or resin remaining on the fiber. This variability makes a quantitative deconvolution of functionalities unreliable. For these reasons, while we cannot perform a quantitative analysis as suggested, we have focused on providing qualitative considerations to better explain the observed changes in the recycled samples. However, for a more detailed quantitative analysis of the functionalities, we have included XPS data in the manuscript, which can provide the desired information on the functional groups present in the reference and recycled fibers.
- Additionally, a comparison table with attributes like Tensile Strength obtained for CFs via this approach with other previously reported methods can be interesting to be shown in a table which I believe will increase the impact of the work.
RESPONSE: Thank you for your suggestion. We have included a comparison table (now added as Table 13 in the manuscript) that highlights the tensile strength values obtained for rCFs in previous studies. We also analyze the effect of each process on the properties of the CFs.
- Minor comment
There are several areas in the manuscript where I saw an error message like Error! Reference source not found. I believe this may be because of some error due to referencing software issues. Please rectify this
RESPONSE: Thank you for the comment. We have carefully reviewed all the crosslinks in the text and corrected the errors related to the "Error! Reference source not found." issue. The correct cross-references for tables and figures have now been added throughout the manuscript.

Round 2
Reviewer 1 Report
Comments and Suggestions for Authors
The authors made all corrections that were needed. The paper can be now accepted
Reviewer 2 Report
Comments and Suggestions for Authors
Thanks to the authors for providing the revised version. The manuscript can be accepted in its present form.